# Needle In A Multimodal Haystack

**Weiyun Wang**[1,2], **Shuibo Zhang**[2], **Yiming Ren**[3,2], **Yuchen Duan**[4,2], **Tiantong Li**[3,2],
**Shuo Liu**[2], **Mengkang Hu**[7,2], **Zhe Chen**[5,2], **Kaipeng Zhang**[2], **Lewei Lu**[6], **Xizhou Zhu**[3,2,6],
**Ping Luo**[7,2], **Yu Qiao**[2], **Jifeng Dai**[3,2], **Wenqi Shao**[2✉], **Wenhai Wang**[4,2✉]

[1]Fudan University, [2]OpenGVLab, Shanghai AI Laboratory, [3]Tsinghua University,
[4]The Chinese University of Hong Kong, [5]Nanjing University,
[6]SenseTime Research, [7]The University of Hong Kong

## Abstract

With the rapid advancement of multimodal large language models (MLLMs), their evaluation has become increasingly comprehensive. However, understanding long multimodal content, as a foundational ability for real-world applications, remains underexplored. In this work, we present Needle In A Multimodal Haystack (MM-NIAH), the first benchmark specifically designed to systematically evaluate the capability of existing MLLMs to comprehend long multimodal documents. Our benchmark includes three types of evaluation tasks: multimodal retrieval, counting, and reasoning. In each task, the model is required to answer the questions according to different key information scattered throughout the given multimodal document. Evaluating the leading MLLMs on MM-NIAH, we observe that existing models still have significant room for improvement on these tasks, especially on vision-centric evaluation. We hope this work can provide a platform for further research on long multimodal document comprehension and contribute to the advancement of MLLMs. Code and benchmark are released at https://github.com/OpenGVLab/MM-NIAH.

## 1   Introduction

With the advancements in Large Language Models (LLMs) [1, 2, 3, 4, 5, 6, 7], significant strides have also been made in Multimodal Large Language Models (MLLMs) [8, 9, 10, 11, 12, 13, 14, 15, 16, 17] across various vision-language tasks. Recently, some MLLMs [18, 19, 20, 21, 22, 23, 24] have begun to explore a wider range of applications, from basic dialogue to document-level long context understanding, by leveraging interleaved image-text documents as training corpora. However, due to the limitations of context window size, most existing MLLMs struggle to effectively comprehend long-context multimodal documents. In addition, the lack of appropriate evaluation benchmarks is a key factor that limits the further development of MLLMs for long-context multimodal understanding.

As shown in Fig. 1a, existing benchmarks for multi-image comprehensions, such as SEED-Bench-2 [25] and BLINK [26], consist of short contexts, which fail to evaluate the capability for long-context document comprehension. Additionally, benchmarks for video question answering, like MVBench [27], concentrate on vision-dominant video understanding rather than text-dominant multimodal document understanding (see Fig. 1b). Constructing benchmarks for multimodal long-context comprehension poses several challenges. (1) The lack of high-quality multimodal long-context datasets, which require substantial resources and effort to create; (2) The need for evaluation questions that are sufficiently complex to require models to integrate information from the entire long context

---

✉ Corresponding Authors: wangwenhai@pjlab.org.cn; shaowenqi@pjlab.org.cn

38th Conference on Neural Information Processing Systems (NeurIPS 2024) Track on Datasets and Benchmarks.

This is my daily exercise goal:

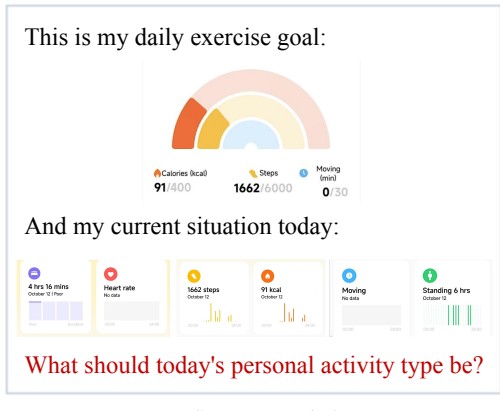

And my current situation today:

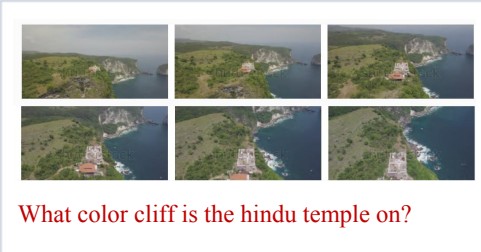

What should today's personal activity type be?

(a) SEED-Bench-2

What color cliff is the hindu temple on?

(b) MVBench

The Denver victory is great for all of us in the living architecture field, but especially for the citizens of Denver. And hopefully it's a sign ...

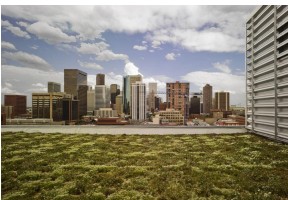

Mayor Hancock was against the measure, saying it went too far. The pirate's treasure contains a compass. Regarding the passage of ...

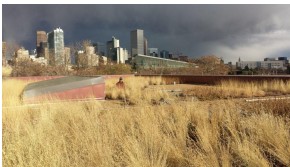

At present, the mayor and city council members are researching all of the possible opportunities and challenges that might be inherent in implementing the initiative. Green Roofs for ...

What does the pirate's treasure contain?

(c) MM-NIAH (ours)

Figure 1: **Comparison of MM-NIAH with other multi-image benchmarks.** Our MM-NIAH focuses on the evaluation of long multimodal document comprehension.

to answer correctly; and (3) The fact that existing multimodal models have not been evaluated on long-context multimodal content, highlighting the necessity for robust evaluation protocols to fairly compare the performance of current methods.

In this work, we introduce MM-NIAH, the first benchmark designed to systematically evaluate the comprehension capability of existing MLLMs for long multimodal documents. As shown in Fig. 1c, MM-NIAH requires the model to answer questions related to the key information scattered throughout the multimodal document. To build this benchmark, we concatenate multiple interleaved image-text documents from OBELICS [24] into a long-context document containing 1k to 72k image and text tokens. After that, we inject needles containing key information into a certain depth of the text or certain images within the document. To cover both text and image modalities, the proposed MM-NIAH comprises two types of needles (*i.e.*, text needles and image needles), where the needles inserted into the text are termed text needles while those inserted into images are termed image needles. For a comprehensive evaluation, we design three types of tasks, including retrieval, counting, and reasoning in our MM-NIAH. The retrieval task requires models to find the key information inserted into the text or images within the document. The counting task contains multiple needles, and the model must collect all needles and count the number of them. The reasoning task asks the model to reason over the cues from multiple needles which are scattered throughout the document.

Based on MM-NIAH, we conduct experiments to evaluate open-source and close-source MLLMs. The experimental results demonstrate that (1) Existing MLLMs perform considerably worse with image needles than with text needles; (2) Existing MLLMs pre-trained on image-text interleaved data do not exhibit superior performance on MM-NIAH compared to those pre-trained only on image-text pair data; (3) MLLMs fail to maintain the long context capability of their underlying LLMs; (4) While RAG enhances performance on text needles, it is ineffective for image needles in the MM-NIAH benchmark. More detailed conclusions and analyses can be found in Section 4.2.

In summary, our main contributions are as follows:

(1) We construct MM-NIAH, the first benchmark designed to systematically evaluate the comprehension capability of existing MLLMs for long multimodal documents, which provides a platform for further research on long multimodal document comprehension.

(2) We extend MLLMs with RAG to serve as a powerful baseline, which greatly enhances the text needles retrieval ability while making trivial improvements for image needles. This demonstrates that the RAG method is unsuitable for our MM-NIAH.

(3) We evaluate the long-context performance of 9 advanced MLLMs on MM-NIAH, where the context length ranges from 1k to 72k. Experimental results reveal that both open-source and closed-source MLLMs struggle to comprehend long multimodal documents accurately, suggesting that long multimodal document comprehension remains a challenging problem.

## 2 Related Work

### 2.1 Multimodal Large Language Models

Multimodal Large Language Models (MLLMs) have achieved impressive performance across various vision-language tasks, enabling large language models to understand the visual world [8, 11, 13, 15, 17, 28, 29, 30, 31, 32]. In the realm of MLLMs, OpenAI introduced GPT-4V [33], extending GPT-4's capabilities to incorporate visual inputs. Google's Gemini series evolved from Gemini 1.0 [34] to Gemini 1.5 [35], enhancing its abilities to process text, images, and audio data. There are also open-sourced MLLMs [15, 18, 19, 20, 36, 37, 38, 39, 40, 41, 42, 43, 44, 45] which has greatly promoted the development of the field. Well-known examples include: BLIP series [9, 10, 46], LLaVA series [11, 12, 47], VisionLLM [8], Qwen-VL [13], All-Seeing series [14, 15], and others [16, 17, 48, 49, 50, 51]. However, existing MLLMs are constrained by a limited context window size, impeding their ability to comprehend long multimodal documents. For instance, Emu2 [21] can handle a maximum of 2048 tokens, while InternVL-1.5 [17] can process up to 4096 tokens. This constraint reveals that long-context multimodal understanding remains a significant challenge.

### 2.2 Multimodal Benchmarks

The rapid advancements in Multimodal Large Language Models (MLLMs) have led to the development of various benchmarks designed to comprehensively assess their multimodal reasoning capabilities. Early benchmarks focused on single tasks [15, 52, 53, 54, 55, 56, 57, 58, 59, 60, 61]. For example, DocVQA [60] is designed for OCR-centric evaluation, POPE [61] is designed for hallucination evaluation, and CRPE [15] is designed for relation comprehension evaluation. Recently, a series of efforts have shifted towards more holistic evaluations. Benchmarks such as MME [62], LVLM-eHub [63], SEED-Series [64, 65], MM-Vet [66], MMBench [67], MMT-Bench [68], MMMU [69] and others [27, 70, 71, 72, 73, 74, 75] have attempted to provide a broader assessment of the reasoning abilities across multiple tasks and modalities of MLLMs. However, these benchmarks are still limited to relatively short contexts, typically consisting of a single image or a short sequence of images and text. Besides, despite the long context introduced by numerous frames, benchmarks designed for long video qa [27, 76, 77, 78, 79] concentrate on vision-dominant video understanding rather than text-dominant multimodal document understanding. Therefore, evaluating the ability to understand long multimodal documents remains an underexplored problem. In this work, we propose MM-NIAH to evaluate the comprehension capability of existing MLLMs for long multimodal documents.

### 2.3 Needle In A Haystack

The Needle-In-A-Haystack (NIAH) test is a classic method in natural language processing used to evaluate the ability to understand long context. The vanilla NIAH benchmark [80] introduces a retrieval task where the model is required to retrieve short text (needle) from a long document (haystack). The subsequent works propose a series of more complex tasks, inserting needles containing more information into the documents. For example, BABILong [81] is built upon the reasoning QA from the bAbI[82] dataset, creating a reasoning-related NIAH benchmark. Experimental results on BABILong demonstrate that Retrieval Augmented Generation (RAG) has no positive impact on reasoning tasks. Counting Stars [83] requires the model to collect inter-dependency across multiple pieces of evidence spanning the entire context and summarize them into a specified answer. The recent RULER benchmark [84] introduces four different tasks, including retrieval, multi-hop tracing, aggregation, and question answering, to evaluate the long-context capability from multiple perspectives. These benchmarks are all text-only and struggle to evaluate the long-context understanding

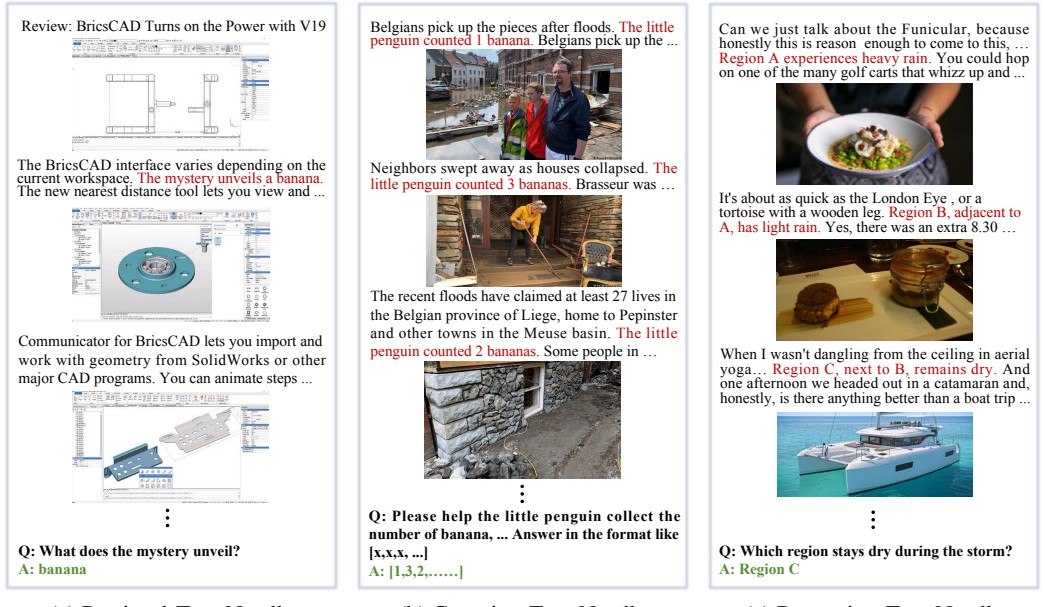

(a) Retrieval-Text-Needle     (b) Counting-Text-Needle     (c) Reasoning-Text-Needle

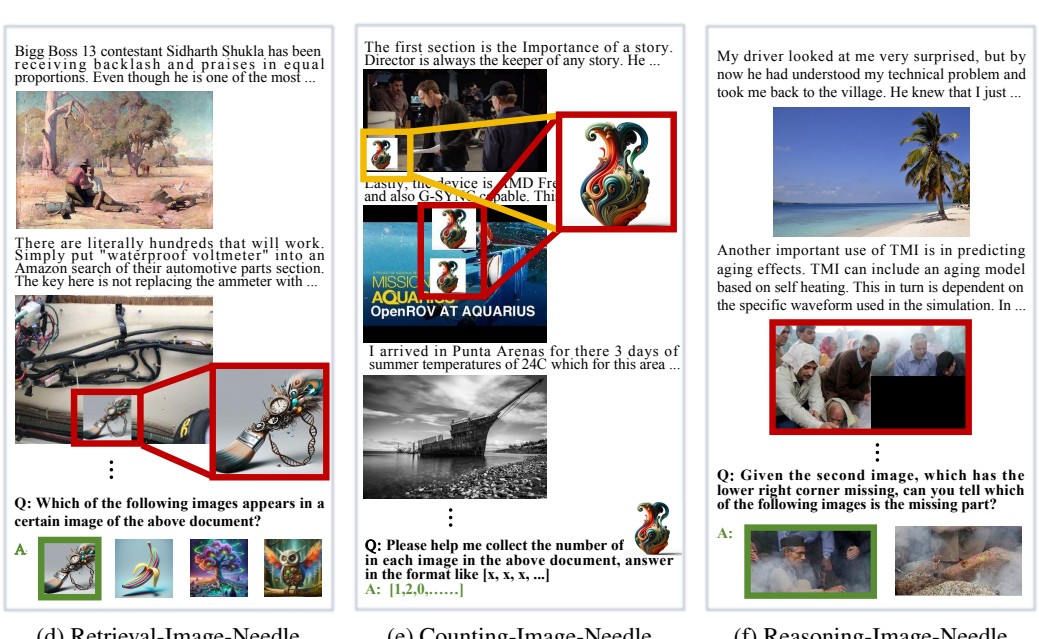

(d) Retrieval-Image-Needle     (e) Counting-Image-Needle     (f) Reasoning-Image-Needle

Figure 2: **Examples for each task in MM-NIAH.** Our MM-NIAH consists of three tasks and two types of needles, formulating six types of evaluation data in total. Note that Retrieval-Image-Needle and Reasoning-Image-Needle are formulated as single-choice questions.

ability of MLLMs. Our MM-NIAH is the first multimodal NIAH benchmark designed to evaluate the comprehension ability for long multimodal documents.

## 3 Needle In A Multimodal Haystack

In this section, we introduce Needle In A Multimodal Haystack (MM-NIAH), a benchmark designed to systematically evaluate the comprehension ability for long multimodal documents. This benchmark requires the model to answer specific questions according to the key information scattered throughout the multimodal document. To generate the evaluation data, we first concatenate interleaved image-

text sequences from OBELICS [24] to establish the background documents, termed "multimodal haystacks". Then, we generate three data types based on these documents: retrieval, counting, and reasoning. We insert either text needles or image needles into documents for each task.

### 3.1 Multimodal Haystack

Due to the absence of open-source long multimodal document data, we concatenate the interleaved image-text sequences from OBELICS [24] to establish the multimodal haystack. The tiktoken [1] is utilized to compute the number of text tokens. For the computation of image tokens, we argue that the image should be considered in the statistics of the context length of a given multimodal document. Besides, images with different resolutions should correspond to different numbers of image tokens, as humans expend varying amounts of effort to understand the information in images of different sizes. Therefore, we use the same method as InternVL-1.5 [16] to split the image into several fixed-size patches while maintaining the aspect ratio as much as possible. Each patch is considered to be 256 image tokens. To ensure that the generated document is not dominated by numerous images, we control the concatenation process so that about every 2k text tokens include one image.

### 3.2 Multimodal Needle

The evaluation data in MM-NIAH consists of three tasks: retrieval, counting, and reasoning. The needles are inserted into either text or images in the documents. Those inserted into text are termed text needles, whereas those within images are referred to as image needles. All text needles used in MM-NIAH are manually designed. To keep simplicity, each document contains only one type of needle. The data examples for each task are shown in Fig. 2.

**Retrieval.** The text needle in the retrieval task is a random fact or statement inserted into a certain document depth. The corresponding question asks the model to retrieve this statement. The image needle is a random cartoon-style image generated by DALLE-3 [85], which is inserted into a certain image within the document, and the corresponding question is formulated as a single-choice question. The model is asked to select the image that appears in the document among four image options.

**Counting.** The text needle in the counting task comprises a series of statements, each of which claims the little penguin counted a certain number of needles. For the image needles, a certain number of cartoon-style images are inserted into each image within the document, serving as the needles to be counted. Inspired by the Counting Stars benchmark [83], we require the model to list the number of needles in each statement or image instead of directly outputting the total number of needles. The motivation behind this design is to ensure that the model accurately retrieves and comprehends all text and image needles inserted into the multimodal document.

**Reasoning.** A series of statements are inserted into different positions of the given document to serve as the text needle. The model must retrieve all these statements and reason over them to answer the question correctly. Besides, for each evaluation data, images sampled from the Jigsaw and Multi-view reasoning split of BLINK benchmark [26] are inserted into the document to serve as the image needle. The model is required to answer the question related to these images.

Based on the above design, MM-NIAH comprises six types of data in total, each containing approximately 3,000 samples. We generate 40 different needles for text retrieval, 12 for text counting, 57 for text reasoning, 14 for image retrieval and image counting, and 288 for image reasoning. We place these needles into different positions within various documents to create different evaluation samples. When generating evaluation samples, we carefully control the distribution of context length and needle depth to ensure they are as uniform as possible. For visualization in Section 4, each slot in our heatmaps (see Fig. 3) contains around 50 samples to ensure the stability of the evaluation.

### 3.3 Data Statistics

The data statistics of MM-NIAH are presented in Tab. 1, which summarizes the answer type, number of data samples, and needles inserted into the multimodal haystack for each task. Our benchmark comprises about 12k samples in total. For the multimodal haystack, we limit the maximum number of tokens to 72k with at most 36 images. The number of text needles denotes the number of statements

---

[1]https://github.com/openai/tiktoken

Table 1: **Data statistics of MM-NIAH.** "#" denotes the number of something.

| Task | Needle Type | Answer Type | #Samples | #Needles Per Sample |
|---|---|---|---|---|
| Retrieval | Text | Open-Ended | 2798 | 1 |
| | Image | Multi-Choice | 2782 | 1 |
| Counting | Text | Open-Ended | 2828 | 1$\sim$3 |
| | Image | Open-Ended | 2532 | 1$\sim$5 |
| Reasoning | Text | Open-Ended | 2774 | 3 |
| | Image | Multi-Choice | 2772 | 1$\sim$2 |

inserted into the multimodal haystack, while the number of image needles denotes the number of images, which are pasted with a cartoon-style image generated by DALLE-3 [85] or sampled from BLINK [26], within the document. For the counting task with image needles, even though at most 5 images can be pasted with cartoon-style images, we still require the model to output a list enumerating the number of needles in each image of the document. We argue that this formulation requires the model to understand the details of all images within the document in order to achieve good performance on this task.

### 3.4 An Improved Baseline with Retrieval Augmented Generation.

We augment InternVL-1.5 [17] with Retrieval Augmented Generation (RAG) as a stronger baseline. Each sample in MM-NIAH consists of a multimodal document and a question-answer pair. Given the multimodal document, we first retrieve a portion of this document conditioned on this question and then ask the model to answer the question based on the retrieved portion instead of the entire document. Specifically, each multimodal document is represented as an interleaved image-text sequence $x = (x_1, x_2, ..., x_n)$, where $x_i$ can be a text sentence or an image. The question $q$ and text sentences are encoded by the text encoder of InternVL-G [16], while the images are encoded by the image encoder of InternVL-G. Note that we encode each sentence separately. Subsequently, we obtain the similarity sequence $s = (s_1, s_2, ..., s_n)$, where $s_i$ denotes the cosine similarity between the embeddings of $q$ and $x_i$. The retrieved portion consists of those $x_i$ with the highest $s_i$, maintaining the relative order within $x$ and ensuring that the number of retrieved tokens is smaller than the pre-defined length limit. We compute the number of image tokens using the method introduced in Section 3.1.

## 4 Experiments

### 4.1 Experimental Settings

**Baselines.** We evaluate six leading open-source MLLMs and two leading closed-source MLLMs on our MM-NIAH. Among the open-source MLLMs, we consider LLaVA-1.6 [12], InternVL-1.5 [17], VILA [86], Emu2-Chat [21], and IDEFICS2 [87] as our baselines. Among these models, LLaVA-1.6 and InternVL-1.5 are trained on image-text pair data without using image-text interleaved data, while VILA, Emu2-Chat, and IDEFICS2 are trained with image-text interleaved data. Note that the training corpora of IDEFICS2 include OBELICS [24]. For the closed-source MLLMs, we consider Gemini-1.5-Flash [35] and GPT-4V [33] as baseline models. Due to the constraint that the API of GPT-4V only supports up to 10 images, we evaluate GPT-4V only on our text-needle data. Human performance is also provided as a baseline, which is obtained by asking 10 human experts to each complete a portion of the evaluation data and then merging all the results. They are allowed to use the "Find" operation of the browser during the evaluation process.

**Metrics.** For the retrieval and reasoning tasks, we utilize Accuracy as the evaluation metric, implemented based on the LVLM-eHub [88]. For the counting task, we use Soft Accuracy, defined as $\frac{1}{N} \sum_{i=1}^{N} \frac{m_i}{M_i}$, where $m_i$ is the number of matched elements in the corresponding positions between the predicted and ground-truth lists and $M_i$ is the number of elements in the ground-truth list for the $i$-th sample. Note that the required output for this task is a list.

**Evaluation.** We evaluate all open-source MLLMs based on the transformers library [89]. During the evaluation process, we evenly split each model into 8 A100 GPUs and use the Flash Attention [90, 91]

to save memory usage. We do not truncate the context and directly input the entire document to these models even if the context length is larger than the max length during their training process.

## 4.2 Comparison of Advanced MLLMs on MM-NIAH

In this section, we present the evaluation results in heatmap format (see Fig. 3). In the heatmaps, green slots indicate higher performance, while red slots indicate lower performance. The x-axis in Fig. 3 represents context length. We divided the context length into different bins, which form the x-axis of the heatmap. For example, the slot in the top left corner represents accuracy when the given context length ranges from 1K to 2K and the needle depth ranges from 0 to 0.2. Additionally, the average performance across depths for each context length range is presented in table format (see Appendix A). The main findings from these results are detailed as follows.

**Performance degrades while context length increases.** As illustrated in Fig. 3, there is a noticeable decline in model performance as the context length increases. This trend can be observed across all evaluated models and tasks. The degradation is more evident in tasks requiring higher levels of understanding and reasoning, indicating that current models struggle to maintain accuracy when dealing with longer multimodal documents. We also find that contemporary open-source MLLMs can not follow the instructions and begin to produce gibberish when the context is quite lengthy.

**Image needles are much more difficult than text needles.** The results show that models exhibit significantly weaker comprehension capabilities for image needles than text needles. This gap is evident across all tasks, where the performance for image needles remains considerably lower than that for text needles across all models. Although the retrieval and reasoning tasks are formulated as single-choice questions, we find that MLLMs fail to understand the image choices and tend to produce gibberish when the context is lengthy. As a result, the performance may be even worse than random guessing. Besides, in the counting task, we qualitatively observe that the lengths of the predicted list always mismatch with the number of images within the documents. This phenomenon demonstrates that existing MLLMs even struggle to recognize the exact number of images within the documents, suggesting the poor image comprehension ability for long multimodal documents.

**Models pre-trained on image-text interleaved data do not exhibit superior performance.** Experimental results show that models like VILA [86] and Emu2-Chat [21], which are trained with image-text interleaved data, do not exhibit substantial performance improvements over models only trained on image-text pairs, such as LLaVA-1.6 [12] and InternVL-1.5 [17]. This indicates that simply training on interleaved data is insufficient for improving long multimodal document understanding, suggesting that alternative approaches or more sophisticated training techniques are necessary.

**The "Lost in the Middle" problem also exists in MLLMs.** The "Lost in the Middle" problem is widely recognized for LLMs [92], where models perform worse on identifying relevant information in the middle sections of documents compared to the beginning and end sections. When evaluating MLLMs with text needles, we can also observe this trend. The end sections typically show the best performance, followed by the beginning sections, with the middle sections performing the worst. Although this phenomenon is not evident for image needles, we believe this is because the model's overall ability to understand images in multimodal documents is weak, thus not reflecting this trend.

**The most advanced MLLM still struggles to comprehend multimodal documents.** Even Gemini-1.5 [35], one of the most advanced multimodal models, fails to achieve ideal performance in our MM-NIAH. Notably, the performance in image needles of Gemini-1.5 is also quite poor, with a significant gap compared to human performance. This indicates that there is still significant room for improvement in multimodal document comprehension.

**Long context capability of LLMs is NOT retained in MLLMs.** Despite the powerful ability of open-source LLMs to handle very long context window size (*e.g.*, Yi-34B [93] and InternLM2-20B [5] with 200K tokens), this capability does not fully transfer to MLLMs. For instance, InternVL-1.5, which is built upon InternLM2, exhibits a decline in performance when dealing with contexts longer than 32k tokens. In contrast, InternLM2 can nearly perfectly find needles in a 200k-long context. Therefore, we believe that enhancing the robustness of MLLMs to maintain high performance across extended contexts remains a critical research direction.

**RAG boosts Text Needle Retrieval but not Image Needle Retrieval.** RAG significantly enhances the capability of InternVL-1.5 in retrieving text needles from long multimodal documents compared

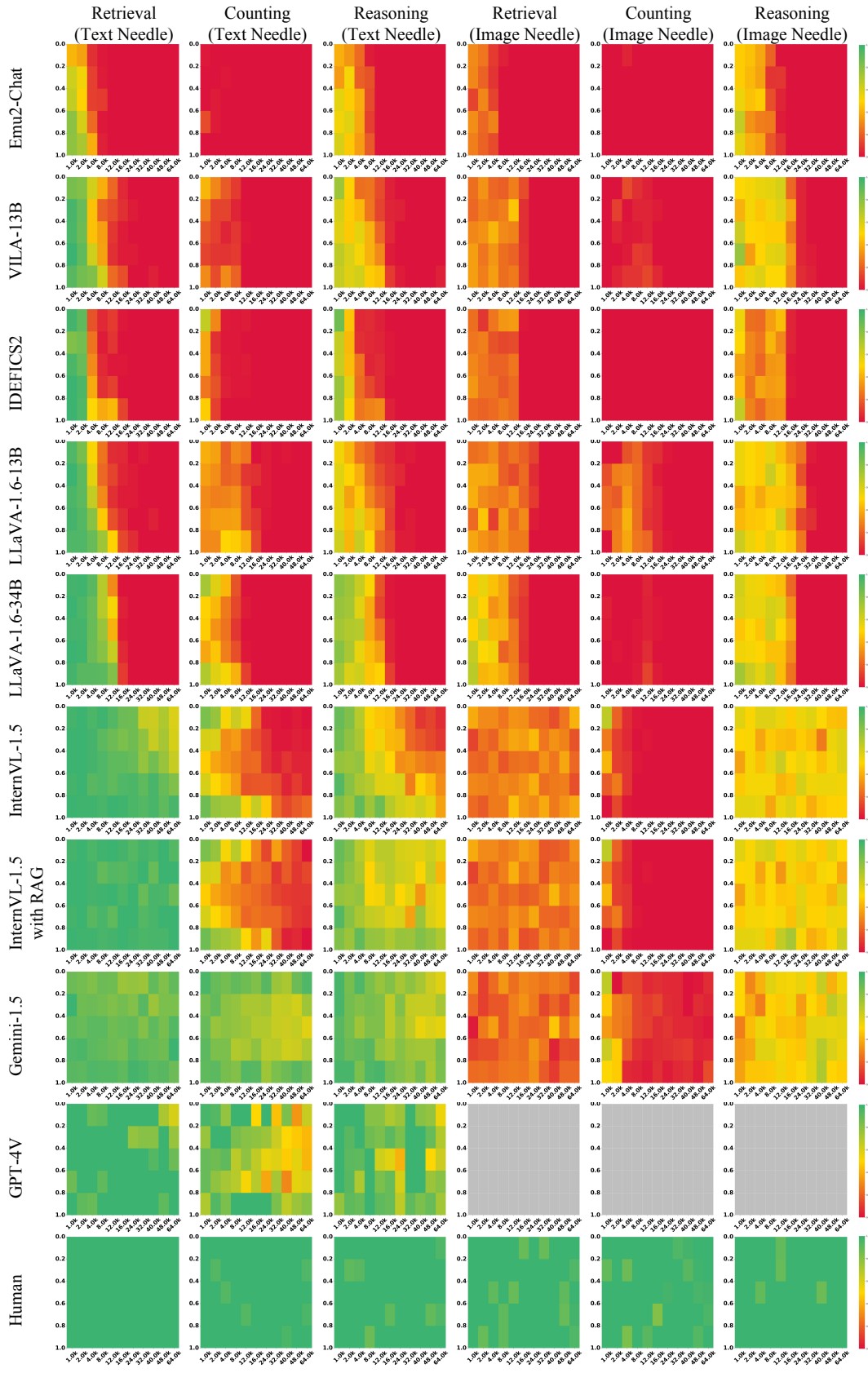

Figure 3: **Results on MM-NIAH.** Green slots indicate higher performance, while red slots indicate lower performance. We evaluate GPT-4V only on our text-needle data because of the constraint that the API of GPT-4V only supports up to 10 images.

to the counterpart without RAG. However, this enhancement does not extend to the retrieval of image needles, where the performance remains poor. For the retrieval and reasoning task, the RAG method might fail to keep the image where the image needle is inserted in the extracted chunks. For the counting task, we require the model to output the number of image needles inserted into each image in the document. Therefore the model has to comprehend all images accurately. However, since the RAG method only extracts a portion of the document, some images might be omitted when the multimodal document is lengthy, leading to the failure in our proposed counting task. These results and analysis demonstrate that RAG methods are unsuitable for the image needles in MM-NIAH as the benchmark requires a comprehensive understanding of all images within the multimodal document.

**Humans achieve near-perfect performance on MM-NIAH.** As shown in the bottom of Fig. 3, humans achieve near-perfect performance on MM-NIAH, highlighting the gap between human-level comprehension and the abilities of current MLLMs. It is important to note that achieving perfect performance on MM-NIAH does not equate to perfect long document understanding ability. However, it serves as a prerequisite for achieving long multimodal document understanding ability.

**Training on background documents does not boost performance on MM-NIAH.** Since the multimodal haystack in MM-NIAH is obtained by concatenating interleaved image-text sequences from OBELICS [24], a general concern is the potential data contamination for models trained with OBELICS [24]. However, evaluation results of IDEFICS2 [87] on MM-NIAH show that IDEFICS2, compared to other models, does not demonstrate any performance advantage. We attribute this to the fact that while models trained on OBELICS may have a better understanding of these documents, the text and image needles are newly inserted into these documents, and the generated questions are only related to this newly inserted content, effectively avoiding the risk of data contamination.

**MLLMs fail to recognize the exact number of images in the document.** As discussed above, we qualitatively observe that the lengths of the predicted list from MLLMs always mismatch with the number of images within the documents. To analyze this phenomenon quantitatively, we ask Gemini-1.5 [35] to output the number of images contained in the given document and compute the accuracy. The experimental results are depicted in Fig. 4. We can observe that the accuracy decreases as the number of images in the context grows, indicating the poor performance of Gemini-1.5 in recognizing the exact number of images in the document. Notably, when the number of images within the document exceeds five, the accuracy drops below 10%. This

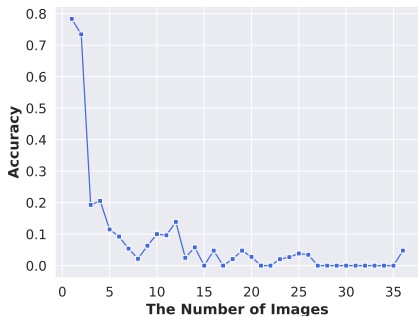

Figure 4: **Accuracy of Gemini-1.5 to output the number of images in context.**

suggests that a major reason for the poor model performance in tasks with image needles is that current models fail to recognize the exact number of images in the document.

## 5    Conclusion & Limitation

In this work, we propose MM-NIAH, the first benchmark designed to systematically evaluate the comprehension ability for long multimodal documents. MM-NIAH comprises three types of evaluation tasks: multimodal retrieval, counting, and reasoning. The model is required to answer specific questions according to the key information scattered throughout the given multimodal document. Evaluating the leading MLLMs on MM-NIAH, we observe that existing models still have significant room for improvement on these tasks, especially on vision-centric evaluation. We also demonstrate that the RAG method is unsuitable for the image needles in MM-NIAH, suggesting that the long multimodal document comprehension remains a non-trivial problem for MLLMs.

Regarding limitations, the long multimodal documents in MM-NIAH only serve as the background, and the answer only relates to the needles inserted into it. The construction of evaluation data related to the entire document will leave for future work.

**Broader Impact.** We hope this work can provide a platform for further research on long multimodal document comprehension and contribute to the advancement of MLLMs. We do not foresee obvious undesirable ethical/social impacts at this moment.

## Acknowledgments

The work is supported by the National Key R&D Program of China (NO. 2022ZD0161000 and 2022ZD0161300), the General Research Fund of Hong Kong (No.17200622 and 17209324), and the National Natural Science Foundation of China (Grant No. 62376134).

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

# A More Results

In this section, we present experimental results in table format. The overall performance in MM-NIAH is shown in Tab. 2, which is obtained by averaging the performance across the six tasks in MM-NIAH. We also provide the performance of each task in Tab. 6 to Tab. 11. The performance for each context length range is obtained by averaging the accuracy of that context length range across different needle depths. For samples containing multiple needles, we average the depths of each needle to serve as the needle depth of this sample.

Table 2: **Overall performance on MM-NIAH for each context length range.**

| Model | 1K | 2K | 4K | 8K | 12K | 16K | 24K | 32K | 40K | 48K | 64K | Overall |
|---|---|---|---|---|---|---|---|---|---|---|---|---|
| Emu2-Chat | 33.0 | 27.8 | 17.2 | 5.9 | 0.9 | 0.0 | 0.0 | 0.0 | 0.0 | 0.0 | 0.0 | 7.7 |
| VILA-13B | 44.7 | 39.3 | 34.9 | 28.3 | 22.0 | 8.9 | 1.1 | 0.2 | 0.1 | 0.0 | 0.1 | 16.3 |
| IDEFICS2 | 48.0 | 33.8 | 16.4 | 13.8 | 14.3 | 1.2 | 0.0 | 0.0 | 0.0 | 0.0 | 0.0 | 11.6 |
| LLaVA-1.6-13B | 47.0 | 45.0 | 41.6 | 35.0 | 24.3 | 15.5 | 5.7 | 0.8 | 0.2 | 0.1 | 0.0 | 19.6 |
| LLaVA-1.6-34B | 57.9 | 53.5 | 47.1 | 38.6 | 27.0 | 8.2 | 0.0 | 0.0 | 0.0 | 0.0 | 0.0 | 21.1 |
| InternVL-1.5 | 59.5 | 55.3 | 50.1 | 46.4 | 45.2 | 41.9 | 39.5 | 33.2 | 31.6 | 33.2 | 30.1 | 42.4 |
| InternVL-1.5-RAG | 67.5 | 61.1 | 53.3 | 51.2 | 50.6 | 51.5 | 46.2 | 46.2 | 43.8 | 40.1 | 39.0 | 50.1 |
| Gemini-1.5 | 64.7 | 58.3 | 56.8 | 57.1 | 55.4 | 53.7 | 53.6 | 51.9 | 52.5 | 50.7 | 53.6 | 55.3 |
| GPT-4V | - | - | - | - | - | - | - | - | - | - | - | - |
| Human | 99.7 | 99.1 | 97.9 | 99.0 | 98.5 | 98.8 | 99.9 | 99.4 | 99.2 | 98.6 | 98.5 | 98.9 |

## A.1 More findings

In addition to the findings discussed in Section 4.2, we provide more findings here.

**Placing questions before context does NOT improve model performance.** As shown in Fig. 3, all models perform poorly in understanding image needles, which we attribute to the fact that models struggle to remember the details of each image in a long multimodal document. An intuitive improvement method is placing the question before the context, which allows the model to see the options first and then read the document. However, as illustrated by the error cases (see the first row in Fig. 5), this approach cause models like InternVL-1.5 to fail in following the instructions in the questions. In fact, we observe that this phenomenon holds for all MLLMs, resulting in near-zero performance. Therefore, we do not provide quantitative results but qualitatively analyzed this issue.

**The long context understanding ability of Gemini-1.5 is not perfect.** As claimed by Gemini-1.5 [35], their model achieves near-perfect performance in long context evaluation with video haystack. However, in our benchmark, their model still performed poorly. Notably, in the video haystack, they only insert textual information by overlaying the text "The secret word is "needle"" on a single randomly sampled video frame in a 10.5 hour video. In contrast, in our benchmark, we inserted another image as additional visual information into the images. Furthermore, their video haystack consists of long videos, whereas our multimodal haystack comprises long multimodal documents. These differences both contribute to the performance decline of Gemini-1.5 in our benchmark.

**Multimodal fine-tuning without long context data impairs model's ability to handle long context.** We provide the comparison of InternLM2-20B and InternVL-1.5 in Tab. 3, 4, and 5. The evaluation of InternLM2-20B is conducted based on their official codebase. We omit the images within the context and only evaluate InternLM2-20B on tasks with text needles. Note that InternLM2-20B is the language model used to initialize InternVL-1.5. According to the results in the tables, we can observe that InternLM2-20B and InternVL-1.5 achieve comparable performance when the context length is short. However, when the context length is larger than 32K, the performance of InternVL-1.5 is much inferior to InternLM2-20B, demonstrating that using only samples with a maximum context length of less than 4096 for multimodal fine-tuning can impair the model's ability to handle long contexts. It is worth noting that InternLM2-20B also performs poorly in counting and reasoning tasks, which are more complex than retrieval. This phenomenon is consistent with the conclusions presented in RULER [84], which argues that despite achieving nearly perfect performance on the vanilla NIAH test, almost all models exhibit large degradation on more complex tasks as sequence length increases.

## A.2 Qualitative Analyses

In this section, we present some error cases of InternVL-1.5 [17] in Fig. 5. As discussed in Appendix A.1, the examples in the first row show that placing questions before context fails to improve

Table 3: **Comparison of InternLM2 and InternVL-1.5 in text retrieval task.**

| Model | 1∼2K | 2∼4K | 4∼8K | 8∼12K | 12∼16K | 16∼24K | 24∼32K | 32∼40K | 40∼48K | 48∼64K | 64∼72K |
|---|---|---|---|---|---|---|---|---|---|---|---|
| InternLM2 | 98.4 | 99.4 | 99.2 | 97.3 | 95.3 | 93.1 | 91.9 | 90.7 | 88.2 | 82.4 | 82.7 |
| InternVL-1.5 | 99.0 | 99.7 | 96.3 | 95.1 | 92.3 | 90.9 | 90.6 | 81.0 | 81.3 | 79.7 | 72.7 |

Table 4: **Comparison of InternLM2 and InternVL-1.5 in text counting task.**

| Model | 1∼2K | 2∼4K | 4∼8K | 8∼12K | 12∼16K | 16∼24K | 24∼32K | 32∼40K | 40∼48K | 48∼64K | 64∼72K |
|---|---|---|---|---|---|---|---|---|---|---|---|
| InternLM2 | 79.5 | 68.0 | 58.2 | 46.7 | 38.6 | 32.0 | 22.4 | 12.0 | 9.8 | 8.8 | 5.0 |
| InternVL-1.5 | 67.6 | 60.0 | 46.7 | 46.8 | 33.3 | 28.0 | 17.0 | 8.3 | 5.4 | 7.7 | 6.8 |

Table 5: **Comparison of InternLM2 and InternVL-1.5 in text reasoning task.**

| Model | 1∼2K | 2∼4K | 4∼8K | 8∼12K | 12∼16K | 16∼24K | 24∼32K | 32∼40K | 40∼48K | 48∼64K | 64∼72K |
|---|---|---|---|---|---|---|---|---|---|---|---|
| InternLM2 | 88.1 | 83.0 | 81.0 | 68.7 | 69.0 | 60.9 | 50.6 | 39.4 | 40.1 | 34.0 | 31.9 |
| InternVL-1.5 | 85.6 | 78.3 | 75.7 | 59.3 | 60.6 | 52.1 | 44.9 | 32.4 | 33.3 | 29.9 | 22.3 |

model performance but leading to the model collapse of InternVL-1.5 (*i.e.*, generating responses unrelated to the given question). Additionally, the examples in the second row demonstrate that InternVL-1.5 easily steps into a state of model collapse in the counting task, which means that the model tends to output a list that meets format requirements but lacks meaningful content, rather than accurately counting the number of needles in the text or images within the context. This phenomenon becomes more common when the given multimodal documents are exceptionally lengthy, causing the model to produce nonsensical text instead of answering the questions. Furthermore, the third row of Fig. 5 shows some error cases in the retrieval and reasoning tasks, where InternVL-1.5 struggles to understand the details of the images within the given context to answer these questions correctly. Based on these phenomena, we believe that increasing the model's instruction-following ability, improving its capacity to handle multiple images (so that it can at least accurately understand the number of images contained within the document), and enhancing its perception of image details are necessary steps to improve the model's ability for long multimodal document comprehension.

## B  Ethical discussion

Our benchmark, Needle In A Multimodal Haystack (MM-NIAH), builds upon the OBELICS dataset, which has undergone extensive ethical review and content filtering to ensure compliance with ethical standards. The creation of OBELICS was guided by ethical principles, including respect for content creators' consent decisions and significant efforts to filter inappropriate content, such as pornographic material. Based on this solid foundation, all new contents (*i.e.*, text and image needles) introduced in MM-NIAH are carefully designed and manually verified, ensuring that the benchmark aligns with ethical guidelines and avoids the inclusion of any unreasonable or harmful content.

## C  License and Author Statement

We release the benchmark under the CC-BY license and Terms of Use. It is required to disclose the utilization when this benchmark is used for model evaluation purposes. This license does not replace the licenses of the source materials, and any use of content included in the dataset must comply with the original licenses and applicable rights of its data subjects. The purpose of this statement is to clarify the responsibilities and liabilities associated with the utilization of this benchmark. While we have spared no effort to ensure accuracy and legality of samples in our benchmark, we cannot guarantee its absolute completeness or correctness. Therefore, if any rights, legal or otherwise, are violated through this benchmark, including but not limited to copyright infringement, privacy violations, or misuse of sensitive information, we, the authors, assume no liability for such violations.

By accessing, downloading, or using this benchmark, you agree to assume sole responsibility for any legal or other consequences resulting from its utilization. You also acknowledge your commitment to adhere to all relevant laws, regulations, and ethical guidelines governing its usage. Your acceptance of this statement and adherence to the terms and conditions of the CC-BY license are implicit in your access, download, or utilization of this benchmark. If you do not agree with the terms outlined herein or the CC-BY license, you are not authorized to use this benchmark.

The benchmark will be hosted and maintained on Github and the Hugging Face Hub platform.

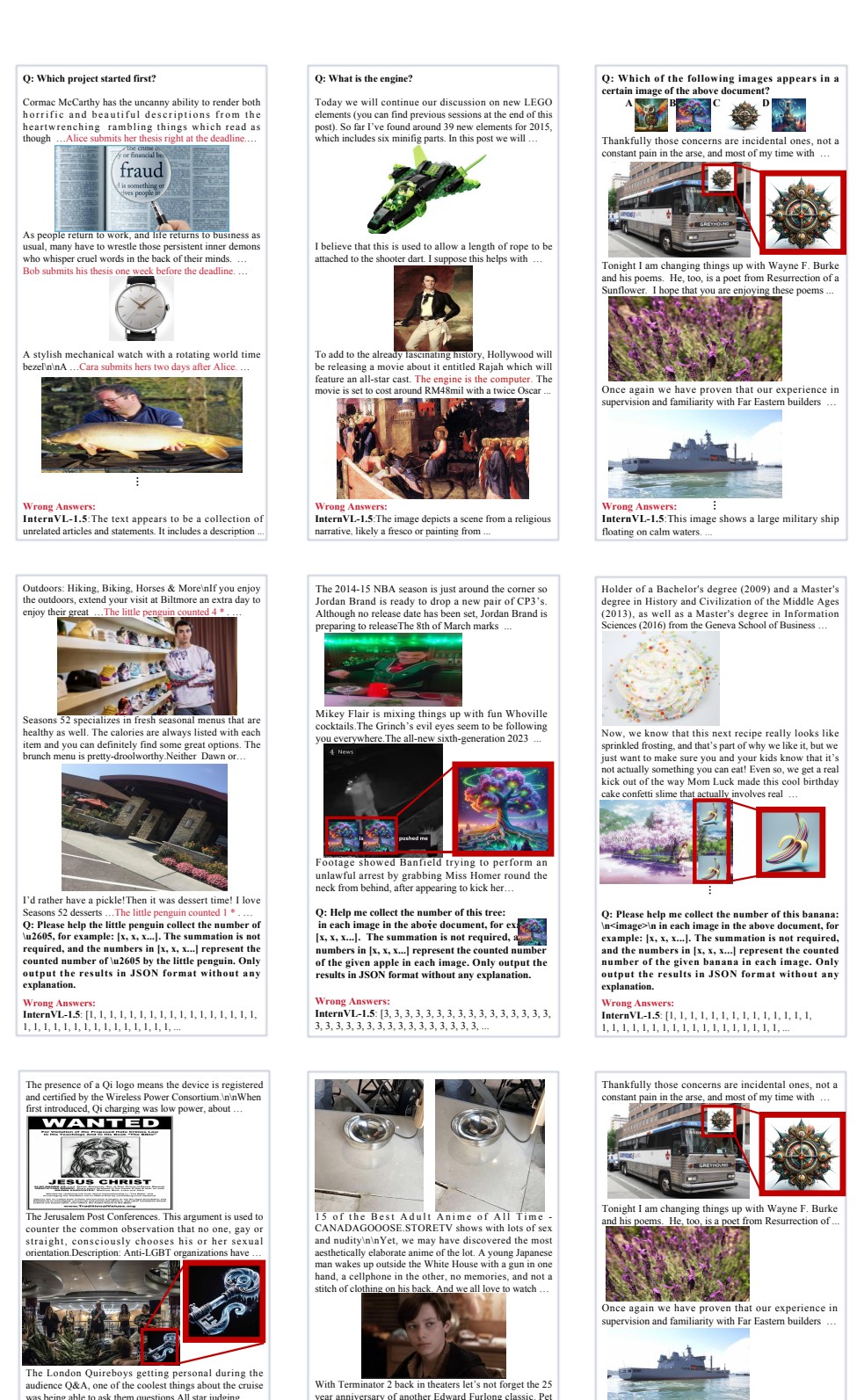

Figure 5: **Some error cases of InternVL-1.5.**

Table 6: **Results on Retrieval-Text-Needle.**

| Model | 1K | 2K | 4K | 8K | 12K | 16K | 24K | 32K | 40K | 48K | 64K | Overall |
|---|---|---|---|---|---|---|---|---|---|---|---|---|
| Emu2-Chat | 65.3 | 54.3 | 18.6 | 3.9 | 0.0 | 0.0 | 0.0 | 0.0 | 0.0 | 0.0 | 0.0 | 12.9 |
| VILA-13B | 93.7 | 86.6 | 59.2 | 38.5 | 15.2 | 6.8 | 0.9 | 0.0 | 0.7 | 0.0 | 0.0 | 27.4 |
| IDEFICS2 | 95.0 | 90.7 | 31.8 | 11.8 | 15.1 | 3.0 | 0.0 | 0.0 | 0.0 | 0.0 | 0.0 | 22.5 |
| LLaVA-1.6-13B | 96.4 | 91.0 | 68.4 | 39.2 | 18.3 | 6.8 | 2.3 | 0.4 | 0.6 | 0.0 | 0.0 | 29.4 |
| LLaVA-1.6-34B | 98.5 | 96.5 | 89.9 | 77.3 | 53.8 | 4.3 | 0.0 | 0.0 | 0.0 | 0.0 | 0.0 | 38.2 |
| InternVL-1.5 | 99.0 | 99.7 | 96.3 | 95.1 | 92.3 | 90.9 | 90.6 | 81.0 | 81.3 | 79.7 | 72.7 | 89.0 |
| InternVL-1.5-RAG | 99.4 | 99.6 | 99.1 | 99.0 | 98.0 | 96.5 | 96.3 | 96.1 | 94.1 | 95.3 | 94.9 | 97.1 |
| Gemini-1.5 | 92.8 | 89.6 | 89.2 | 89.5 | 87.3 | 85.0 | 87.9 | 86.8 | 87.1 | 86.0 | 90.7 | 88.4 |
| GPT-4V | 97.5 | 98.2 | 95.6 | 96.0 | 100.0 | 100.0 | 95.6 | 96.0 | 76.0 | 92.5 | 95.0 | 94.8 |
| Human | 100.0 | 100.0 | 100.0 | 100.0 | 100.0 | 100.0 | 100.0 | 100.0 | 100.0 | 100.0 | 100.0 | 100.0 |

Table 7: **Results on Counting-Text-Needle.**

| Model | 1K | 2K | 4K | 8K | 12K | 16K | 24K | 32K | 40K | 48K | 64K | Overall |
|---|---|---|---|---|---|---|---|---|---|---|---|---|
| Emu2-Chat | 3.2 | 0.8 | 0.5 | 0.2 | 0.0 | 0.0 | 0.0 | 0.0 | 0.0 | 0.0 | 0.0 | 0.4 |
| VILA-13B | 25.5 | 15.4 | 14.8 | 11.2 | 0.4 | 0.0 | 0.0 | 0.0 | 0.0 | 0.0 | 0.0 | 6.1 |
| IDEFICS2 | 42.6 | 15.6 | 1.8 | 1.2 | 1.4 | 0.0 | 0.0 | 0.0 | 0.0 | 0.0 | 0.0 | 5.7 |
| LLaVA-1.6-13B | 33.7 | 32.4 | 30.6 | 33.6 | 21.1 | 6.5 | 1.6 | 0.2 | 0.3 | 0.0 | 0.0 | 14.6 |
| LLaVA-1.6-34B | 55.0 | 47.6 | 34.8 | 19.2 | 3.2 | 0.0 | 0.0 | 0.0 | 0.0 | 0.0 | 0.0 | 14.5 |
| InternVL-1.5 | 67.6 | 60.0 | 46.7 | 46.8 | 33.3 | 28.0 | 17.0 | 8.3 | 5.4 | 7.7 | 6.8 | 29.8 |
| InternVL-1.5-RAG | 80.7 | 70.4 | 52.3 | 52.9 | 57.8 | 52.7 | 40.7 | 36.6 | 28.5 | 19.5 | 12.4 | 45.9 |
| Gemini-1.5 | 90.4 | 85.9 | 82.5 | 79.0 | 79.5 | 79.1 | 75.4 | 71.2 | 70.1 | 74.1 | 77.0 | 78.6 |
| GPT-4V | 70.0 | 90.4 | 84.7 | 84.1 | 82.2 | 72.8 | 73.6 | 64.6 | 55.6 | 53.6 | 77.6 | 73.6 |
| Human | 100.0 | 98.7 | 98.7 | 100.0 | 98.7 | 100.0 | 100.0 | 100.0 | 99.0 | 100.0 | 97.9 | 99.4 |

Table 8: **Results on Reasoning-Text-Needle.**

| Model | 1K | 2K | 4K | 8K | 12K | 16K | 24K | 32K | 40K | 48K | 64K | Overall |
|---|---|---|---|---|---|---|---|---|---|---|---|---|
| Emu2-Chat | 48.7 | 47.5 | 31.1 | 12.8 | 0.0 | 0.0 | 0.0 | 0.0 | 0.0 | 0.0 | 0.0 | 12.7 |
| VILA-13B | 64.9 | 51.9 | 47.4 | 35.6 | 24.5 | 5.2 | 1.2 | 0.0 | 0.0 | 0.0 | 0.7 | 21.0 |
| IDEFICS2 | 73.6 | 48.1 | 17.1 | 11.7 | 10.1 | 1.2 | 0.0 | 0.0 | 0.0 | 0.0 | 0.0 | 14.7 |
| LLaVA-1.6-13B | 57.4 | 42.6 | 46.7 | 33.2 | 19.4 | 11.3 | 2.0 | 1.5 | 0.0 | 0.0 | 0.0 | 19.5 |
| LLaVA-1.6-34B | 76.5 | 69.7 | 61.8 | 43.6 | 27.8 | 4.6 | 0.0 | 0.0 | 0.0 | 0.0 | 0.0 | 25.8 |
| InternVL-1.5 | 85.6 | 78.3 | 75.7 | 59.3 | 60.6 | 52.1 | 44.9 | 32.4 | 33.3 | 29.9 | 22.3 | 52.2 |
| InternVL-1.5-RAG | 89.4 | 86.6 | 79.2 | 66.4 | 63.8 | 69.4 | 63.9 | 61.0 | 64.1 | 59.0 | 58.9 | 69.3 |
| Gemini-1.5 | 95.0 | 87.9 | 84.6 | 87.6 | 83.1 | 74.4 | 78.6 | 72.5 | 70.3 | 66.5 | 70.9 | 79.2 |
| GPT-4V | 95.6 | 93.5 | 89.8 | 93.3 | 79.8 | 79.3 | 65.0 | 98.0 | 76.0 | 76.1 | 76.7 | 83.9 |
| Human | 100.0 | 98.0 | 98.4 | 97.7 | 100.0 | 98.4 | 100.0 | 100.0 | 100.0 | 97.5 | 97.7 | 98.9 |

Table 9: **Results on Retrieval-Image-Needle.**

| Model | 1K | 2K | 4K | 8K | 12K | 16K | 24K | 32K | 40K | 48K | 64K | Overall |
|---|---|---|---|---|---|---|---|---|---|---|---|---|
| Emu2-Chat | 26.3 | 23.6 | 14.8 | 0.7 | 0.0 | 0.0 | 0.0 | 0.0 | 0.0 | 0.0 | 0.0 | 5.9 |
| VILA-13B | 28.8 | 29.1 | 31.1 | 24.7 | 29.8 | 9.6 | 0.0 | 0.0 | 0.0 | 0.0 | 0.0 | 13.9 |
| IDEFICS2 | 26.7 | 21.5 | 22.0 | 22.6 | 23.8 | 0.3 | 0.0 | 0.0 | 0.0 | 0.0 | 0.0 | 10.6 |
| LLaVA-1.6-13B | 32.2 | 34.6 | 26.6 | 26.7 | 24.1 | 23.9 | 6.0 | 0.0 | 0.0 | 0.0 | 0.0 | 15.8 |
| LLaVA-1.6-34B | 57.3 | 51.5 | 43.4 | 34.6 | 23.1 | 9.8 | 0.0 | 0.0 | 0.0 | 0.0 | 0.0 | 20.0 |
| InternVL-1.5 | 25.0 | 24.4 | 26.4 | 26.2 | 33.1 | 31.4 | 31.4 | 28.5 | 25.2 | 30.6 | 26.4 | 28.0 |
| InternVL-1.5-RAG | 24.7 | 30.1 | 32.6 | 36.4 | 27.2 | 27.3 | 24.2 | 31.8 | 20.0 | 15.8 | 16.0 | 26.0 |
| Gemini-1.5 | 17.9 | 17.7 | 22.7 | 23.5 | 25.9 | 26.4 | 27.7 | 20.8 | 21.6 | 19.6 | 22.2 | 22.4 |
| Human | 100.0 | 97.8 | 98.0 | 96.4 | 97.8 | 97.8 | 100.0 | 97.8 | 100.0 | 95.8 | 97.3 | 98.1 |

Table 10: **Results on Counting-Image-Needle.**

| Model | 1K | 2K | 4K | 8K | 12K | 16K | 24K | 32K | 40K | 48K | 64K | Overall |
|---|---|---|---|---|---|---|---|---|---|---|---|---|
| Emu2-Chat | 0.0 | 0.0 | 1.1 | 0.2 | 0.0 | 0.0 | 0.0 | 0.0 | 0.0 | 0.0 | 0.0 | 0.1 |
| VILA-13B | 0.0 | 3.9 | 5.6 | 6.7 | 7.1 | 1.9 | 0.0 | 0.0 | 0.0 | 0.0 | 0.0 | 2.3 |
| IDEFICS2 | 0.0 | 0.0 | 0.0 | 0.4 | 0.1 | 0.0 | 0.0 | 0.0 | 0.0 | 0.0 | 0.0 | 0.0 |
| LLaVA-1.6-13B | 12.0 | 20.2 | 31.7 | 23.1 | 12.3 | 5.5 | 1.0 | 0.0 | 0.2 | 0.4 | 0.0 | 9.7 |
| LLaVA-1.6-34B | 1.3 | 0.3 | 0.4 | 1.1 | 6.0 | 1.2 | 0.0 | 0.0 | 0.0 | 0.0 | 0.0 | 0.9 |
| InternVL-1.5 | 30.6 | 16.6 | 6.1 | 0.7 | 0.5 | 0.3 | 0.0 | 0.0 | 0.0 | 0.0 | 0.0 | 5.0 |
| InternVL-1.5-RAG | 44.8 | 21.8 | 4.9 | 1.8 | 0.6 | 0.2 | 0.0 | 0.0 | 0.0 | 0.0 | 0.0 | 6.7 |
| Gemini-1.5 | 52.1 | 29.8 | 17.0 | 10.4 | 6.9 | 8.3 | 6.0 | 6.3 | 5.0 | 3.6 | 6.4 | 13.8 |
| Human | 98.2 | 100.0 | 94.2 | 100.0 | 98.6 | 96.4 | 99.2 | 98.8 | 98.6 | 98.0 | 98.1 | 98.2 |

Table 11: **Results on Reasoning-Image-Needle.**

| Model | 1K | 2K | 4K | 8K | 12K | 16K | 24K | 32K | 40K | 48K | 64K | Overall |
|---|---|---|---|---|---|---|---|---|---|---|---|---|
| Emu2-Chat | 54.3 | 40.9 | 37.2 | 17.6 | 5.1 | 0.0 | 0.0 | 0.0 | 0.0 | 0.0 | 0.0 | 14.1 |
| VILA-13B | 55.6 | 49.0 | 51.4 | 53.1 | 55.1 | 30.0 | 4.8 | 1.1 | 0.0 | 0.0 | 0.0 | 27.3 |
| IDEFICS2 | 49.8 | 27.1 | 25.6 | 35.3 | 35.3 | 2.7 | 0.0 | 0.0 | 0.0 | 0.0 | 0.0 | 16.0 |
| LLaVA-1.6-13B | 50.1 | 49.2 | 45.7 | 54.1 | 50.7 | 39.1 | 21.0 | 2.4 | 0.0 | 0.0 | 0.0 | 28.4 |
| LLaVA-1.6-34B | 58.8 | 55.4 | 52.2 | 55.7 | 48.1 | 29.2 | 0.0 | 0.0 | 0.0 | 0.0 | 0.0 | 27.2 |
| InternVL-1.5 | 49.2 | 52.8 | 49.5 | 50.1 | 51.3 | 48.5 | 53.2 | 48.9 | 44.4 | 51.1 | 52.2 | 50.1 |
| InternVL-1.5-RAG | 65.9 | 58.3 | 51.5 | 50.8 | 56.4 | 62.7 | 52.1 | 51.4 | 55.9 | 51.2 | 52.0 | 55.3 |
| Gemini-1.5 | 39.6 | 38.9 | 45.1 | 52.3 | 49.7 | 49.1 | 45.7 | 53.7 | 60.9 | 54.1 | 54.3 | 49.4 |
| Human | 100.0 | 100.0 | 98.0 | 100.0 | 95.7 | 100.0 | 100.0 | 100.0 | 97.5 | 100.0 | 100.0 | 99.2 |

# D    Datasheet for MM-NIAH benchmark

## D.1    Motivation

Q1  **For what purpose was the dataset created?** Was there a specific task in mind? Was there a specific gap that needed to be filled? Please provide a description.

- MM-NIAH was created to systematically evaluate the capability of existing Multimodal Large Language Models (MLLMs) to comprehend long multimodal documents, which is crucial for real-world applications but remains underexplored.

Q2  **Who created the dataset (e.g., which team, research group) and on behalf of which entity (e.g., company, institution, organization)?**

- This dataset is presented by OpenGVLab of Shanghai AI Laboratory.

Q3  **Who funded the creation of the dataset?** If there is an associated grant, please provide the name of the grantor and the grant name and number.

- This work was sponsored by Shanghai AI Laboratory.

Q4  **Any other comments?**

- No.

## D.2    Composition

Q5  **What do the instances that comprise the dataset represent (e.g., documents, photos, people, countries)?** *Are there multiple types of instances (e.g., movies, users, and ratings; people and interactions between them; nodes and edges)? Please provide a description.*

- Each instance in MM-NIAH represents a long multimodal document composed of interleaved image-text sequences and a corresponding question-anser pair.

Q6  **How many instances are there in total (of each type, if appropriate)?**

- The MM-NIAH benchmark comprises about 12,000 samples in total. Each type of evaluation data (retrieval, counting, reasoning) contains approximately 2,800 samples, with an equal distribution between text needles and image needles.

Q7  **Does the dataset contain all possible instances or is it a sample (not necessarily random) of instances from a larger set?** *If the dataset is a sample, then what is the larger set? Is the sample representative of the larger set (e.g., geographic coverage)? If so, please describe how this representativeness was validated/verified. If it is not representative of the larger set, please describe why not (e.g., to cover a more diverse range of instances, because instances were withheld or unavailable).*

- The dataset is created based on the interleaved image-text sequences from the OBELICS dataset. It includes a wide range of long multimodal documents to cover diverse scenarios for the evaluation tasks.

Q8  **What data does each instance consist of?** *"Raw" data (e.g., unprocessed text or images) or features? In either case, please provide a description.*

- Each instance consists of a long multimodal document along with inserted needles (either text or image) for evaluation purposes.

Q9  **Is there a label or target associated with each instance?** *If so, please provide a description.*

- Yes, each instance has associated questions related to the inserted needles, which serve as the targets for the evaluation tasks.

Q10  **Is any information missing from individual instances?** *If so, please provide a description, explaining why this information is missing (e.g., because it was unavailable). This does not include intentionally removed information, but might include, e.g., redacted text.*

- No.

Q11  **Are relationships between individual instances made explicit (e.g., users' movie ratings, social network links)?** *If so, please describe how these relationships are made explicit.*

- No.

Q12 **Are there recommended data splits (e.g., training, development/validation, testing)?** *If so, please provide a description of these splits, explaining the rationale behind them.*

- Yes, we provide validation split and test split. Note that the ground truth of the samples in the test split is not publicly available.

Q13 **Are there any errors, sources of noise, or redundancies in the dataset?** *If so, please provide a description.*

- We have conducted a quality check on this benchmark. However, due to the large volume of data, there may be a very small number of errors or omissions.

Q14 **Is the dataset self-contained, or does it link to or otherwise rely on external resources (e.g., websites, tweets, other datasets)?** *If it links to or relies on external resources, a) are there guarantees that they will exist, and remain constant, over time; b) are there official archival versions of the complete dataset (i.e., including the external resources as they existed at the time the dataset was created); c) are there any restrictions (e.g., licenses, fees) associated with any of the external resources that might apply to a future user? Please provide descriptions of all external resources and any restrictions associated with them, as well as links or other access points, as appropriate.*

- The benchmark is built upon the OBELICS dataset for the interleaved image-text sequences. There are no additional external resources required. The MM-NIAH benchmark includes all necessary data for evaluation purposes.

Q15 **Does the dataset contain data that might be considered confidential (e.g., data that is protected by legal privilege or by doctor–patient confidentiality, data that includes the content of individuals' non-public communications)?** *If so, please provide a description.*

- MM-NIAH is built upon the OBELICS dataset, which has undergone extensive ethical review and content filtering to ensure compliance with ethical standards. The creation of OBELICS was guided by ethical principles, including respect for content creators' consent decisions and significant efforts to filter inappropriate content, such as pornographic material. Based on this solid foundation, all new contents (*i.e.*, text and image needles) introduced in MM-NIAH are carefully designed and manually verified, ensuring that the benchmark aligns with ethical guidelines and avoids the inclusion of any unreasonable or harmful content.

Q16 **Does the dataset contain data that, if viewed directly, might be offensive, insulting, threatening, or might otherwise cause anxiety?** *If so, please describe why.*

- See Q15.

Q17 **Does the dataset relate to people?** *If not, you may skip the remaining questions in this section.*

- People might be found in the images or textual descriptions, but they are not the primary emphasis of the dataset.

Q18 **Does the dataset identify any subpopulations (e.g., by age, gender)?**

- We don't include any indicators of subpopulations as attributes.

Q19 **Is it possible to identify individuals (i.e., one or more natural persons), either directly or indirectly (i.e., in combination with other data) from the dataset?** *If so, please describe how.*

- Yes, it may be possible to identify people using face recognition. We do not provide any such means nor make attempts, but institutions owning large amounts of face identifiers may identify specific people in the dataset. Similarly, people may be identified through the associated text.

Q20 **Does the dataset contain data that might be considered sensitive in any way (e.g., data that reveals racial or ethnic origins, sexual orientations, religious beliefs, political opinions or union memberships, or locations; financial or health data; biometric or genetic data; forms of government identification, such as social security numbers; criminal history)?** *If so, please provide a description.*

- See Q15.

Q21 **Any other comments?**

- No.

## D.3 Collection Process

Q22 **How was the data associated with each instance acquired?** *Was the data directly observable (e.g., raw text, movie ratings), reported by subjects (e.g., survey responses), or indirectly inferred/derived from other data (e.g., part-of-speech tags, model-based guesses for age or language)? If data was reported by subjects or indirectly inferred/derived from other data, was the data validated/verified? If so, please describe how.*

- We concatenate multiple interleaved image-text documents from OBELICS into a long-context document containing 1k to 72k image and text tokens. After that, we inject needles containing key information into a certain depth of the text or certain images within the document. To cover both text and image modalities, the proposed MM-NIAH comprises two types of needles (*i.e.*, text needles and image needles), where the needles inserted into the text are termed text needles while those inserted into images are termed image needles. All text and image needles used in MM-NIAH are manually verified to ensure the correctness.

Q23 **What mechanisms or procedures were used to collect the data (e.g., hardware apparatus or sensor, manual human curation, software program, software API)?** *How were these mechanisms or procedures validated?*

- We only use CPUs to generate these evaluation data. We validate our implementation by manually verifying all needles to be inserted and checking a subset of the generated evaluation data.

Q24 **If the dataset is a sample from a larger set, what was the sampling strategy (e.g., deterministic, probabilistic with specific sampling probabilities)?**

- MM-NIAH is built upon the OBELICS dataset. We sample a subset of interleaved image-text sequences from it to construct the multimodal documents for evaluation purposes.

Q25 **Who was involved in the data collection process (e.g., students, crowdworkers, contractors) and how were they compensated (e.g., how much were crowdworkers paid)?**

- No crowdworkers were used in the curation of the dataset. Authors of this paper enabled its creation for no payment.

Q26 **Over what timeframe was the data collected? Does this timeframe match the creation timeframe of the data associated with the instances (e.g., recent crawl of old news articles)?** *If not, please describe the timeframe in which the data associated with the instances was created.*

- The licensed photos vary in their date taken over a wide range of years up to 2023.

Q27 **Were any ethical review processes conducted (e.g., by an institutional review board)?** *If so, please provide a description of these review processes, including the outcomes, as well as a link or other access point to any supporting documentation.*

- As described in Q15, our benchmark is built up on the OBELICS dataset, which has undergone extensive ethical review and content filtering to ensure compliance with ethical standards. Based on this solid foundation, all new contents (*i.e.*, text and image needles) introduced in MM-NIAH are carefully designed and manually verified, ensuring that the benchmark aligns with ethical guidelines and avoids the inclusion of any unreasonable or harmful content. Therefore, we did not conduct a formal ethical review process via institutional review boards.

Q28 **Does the dataset relate to people?** *If not, you may skip the remaining questions in this section.*

- People might be present in the images and descriptions, although they are not the sole emphasis of the dataset.

Q29 **Did you collect the data from the individuals in question directly, or obtain it via third parties or other sources (e.g., websites)?**

- To collect the data, we concatenate multiple interleaved image-text documents from OBELICS into a long-context document containing 1k to 72k image and text tokens. After that, we inject needles containing key information into a certain depth of the text or certain images within the document. To cover both text and image modalities, the proposed MM-NIAH comprises two types of needles (*i.e.*, text needles and image needles), where the needles inserted into the text are termed text needles while those inserted into images are termed image needles. All needles inserted into documents are manually verified.

Q30 **Were the individuals in question notified about the data collection?** *If so, please describe (or show with screenshots or other information) how notice was provided, and provide a link or other access point to, or otherwise reproduce, the exact language of the notification itself.*

- Individuals were not notified about the data collection. Our benchmark is built upon the OBELICS dataset, which only contains information that is publicly available on the Internet. The publishers of these information are usually aware that it will be made public to the world, but they may not be aware that it will be collected in this way.

Q31 **Did the individuals in question consent to the collection and use of their data?** *If so, please describe (or show with screenshots or other information) how consent was requested and provided, and provide a link or other access point to, or otherwise reproduce, the exact language to which the individuals consented.*

- No. See Q30.

Q32 **If consent was obtained, were the consenting individuals provided with a mechanism to revoke their consent in the future or for certain uses?** *If so, please provide a description, as well as a link or other access point to the mechanism (if appropriate).*

- Users can contact us to remove any annotation in our proposed MM-NIAH.

Q33 **Has an analysis of the potential impact of the dataset and its use on data subjects (e.g., a data protection impact analysis) been conducted?** *If so, please provide a description of this analysis, including the outcomes, as well as a link or other access point to any supporting documentation.*

- No. See Q27.

Q34 **Any other comments?**

- No.

### D.4 Preprocessing, Cleaning, and/or Labeling

Q35 **Was any preprocessing/cleaning/labeling of the data done (e.g., discretization or bucketing, tokenization, part-of-speech tagging, SIFT feature extraction, removal of instances, processing of missing values)?** *If so, please provide a description. If not, you may skip the remainder of the questions in this section.*

- MM-NIAH is established using the method described in Q22. All needles to be inserted into the multimodal documents are manually verified to ensure the correctness. Besides, a subset of evaluation data are sampled to be checked to further ensure the correctness.

Q36 **Was the "raw" data saved in addition to the preprocessed/cleaned/labeled data (e.g., to support unanticipated future uses)?** *If so, please provide a link or other access point to the "raw" data.*

- No.

Q37 **Is the software used to preprocess/clean/label the instances available?** *If so, please provide a link or other access point.*

- No.

Q38 **Any other comments?**

- No.

### D.5 Uses

Q39 **Has the dataset been used for any tasks already?** *If so, please provide a description.*

- Only this paper used this benchmark for experiments up to date.

Q40 **Is there a repository that links to any or all papers or systems that use the dataset?** *If so, please provide a link or other access point.*

- Yes, we will maintain the leaderboard on the project page.

Q41 **What (other) tasks could the dataset be used for?**

- The dataset could be used to evaluate the comprehension ability for long multimodal documents.

Q42 **Is there anything about the composition of the dataset or the way it was collected and preprocessed/cleaned/labeled that might impact future uses?** *For example, is there anything that a future user might need to know to avoid uses that could result in unfair treatment of individuals or groups (e.g., stereotyping, quality of service issues) or other undesirable harms (e.g., financial harms, legal risks) If so, please provide a description. Is there anything a future user could do to mitigate these undesirable harms?*

- No.

Q43 **Are there tasks for which the dataset should not be used?** *If so, please provide a description.*

- Our dataset should only be used for non-commercial academic research.

Q44 **Any other comments?**

- No.

### D.6 Distribution

Q45 **Will the dataset be distributed to third parties outside of the entity (e.g., company, institution, organization) on behalf of which the dataset was created?** *If so, please provide a description.*

- Yes, the benchmark will be open-sourced.

Q46 **How will the dataset be distributed (e.g., tarball on website, API, GitHub)?** *Does the dataset have a digital object identifier (DOI)?*

- The data will be available through GitHub.

Q47 **When will the dataset be distributed?**

- This benchmark is released at `https://github.com/OpenGVLab/MM-NIAH`.

Q48 **Will the dataset be distributed under a copyright or other intellectual property (IP) license, and/or under applicable terms of use (ToU)?** *If so, please describe this license and/or ToU, and provide a link or other access point to, or otherwise reproduce, any relevant licensing terms or ToU, as well as any fees associated with these restrictions.*

- CC-BY-4.0.

Q49 **Have any third parties imposed IP-based or other restrictions on the data associated with the instances?** *If so, please describe these restrictions, and provide a link or other access point to, or otherwise reproduce, any relevant licensing terms, as well as any fees associated with these restrictions.*

- MM-NIAH owns the metadata and release as CC-BY-4.0.
- We do not own the copyright of the images.

Q50 **Do any export controls or other regulatory restrictions apply to the dataset or to individual instances?** *If so, please describe these restrictions, and provide a link or other access point to, or otherwise reproduce, any supporting documentation.*

- No.

Q51 **Any other comments?**

- No.

### D.7 Maintenance

**Q52 Who will be supporting/hosting/maintaining the dataset?**

- Huggingface will support hosting of the metadata.
- OpenGVLab of Shanghai AI Laboratory will maintain the samples distributed.

**Q53 How can the owner/curator/manager of the dataset be contacted (e.g., email address)?**

- https://github.com/OpenGVLab/MM-NIAH

**Q54 Is there an erratum?** *If so, please provide a link or other access point.*

- Not at the moment. We plan to maintain it through GitHub issues and the README file.

**Q55 Will the dataset be updated (e.g., to correct labeling errors, add new instances, delete instances)?** *If so, please describe how often, by whom, and how updates will be communicated to users (e.g., mailing list, GitHub)?*

- No. However, specific samples can be removed on request.

**Q56 If the dataset relates to people, are there applicable limits on the retention of the data associated with the instances (e.g., were individuals in question told that their data would be retained for a fixed period of time and then deleted)?** *If so, please describe these limits and explain how they will be enforced.*

- People may contact us to add specific samples to a blacklist.

**Q57 Will older versions of the dataset continue to be supported/hosted/maintained?** *If so, please describe how. If not, please describe how its obsolescence will be communicated to users.*

- We will only support and maintain the latest version at all times and a new version release of MM-NIAH will automatically deprecate its previous version.

**Q58 If others want to extend/augment/build on/contribute to the dataset, is there a mechanism for them to do so?** *If so, please provide a description. Will these contributions be validated/verified? If so, please describe how. If not, why not? Is there a process for communicating/distributing these contributions to other users? If so, please provide a description.*

- We welcome any contributions to MM-NIAH and we will announce updates regarding dataset extensions on GitHub. However, contributors must demonstrate the quality and harmlessness of the extended data annotations; otherwise, we will not accept these extensions.

**Q59 Any other comments?**

- No.

