# OpenReview forum: "Needle In A Multimodal Haystack"
_NeurIPS.cc/2024/Datasets_and_Benchmarks_Track — NeurIPS 2024 Track Datasets and Benchmarks Poster_

### Official Review · Reviewer_nTjb · 2024-07-25
**Offical Review**

**Rating:** 8
**Confidence:** 4
**Correctness:** Yes
**Clarity:** Yes

**Review:**

Quality: The overall quality of this work is high. The authors have developed a rigorous and well-designed benchmark, conducted extensive experiments, and provided thoughtful analysis of the results.

Clarity: The paper is generally well-written and clearly structured.

Originality: This work presents a novel contribution to the field of multimodal evaluation.

Significance: Substantial significance for the field of multimodal.

Pros:
- First benchmark specifically designed for long multimodal document comprehension
- Innovative approach to evaluating both text and image understanding in a unified framework
- Novel insights into the limitations of current MLLMs

Cons:
- Could benefit from more in-depth error analysis

**Strengths:**

- The MM-NIAH benchmark is the first of its kind designed specifically for assessing long multimodal document comprehension, filling a critical gap in MLLM evaluation.
- The benchmark includes three types of tasks (retrieval, counting, and reasoning) and incorporates both text and image "needles", providing a multi-faceted assessment of model capabilities.

**Additional Feedback:**

No

**Documentation:**

Yes

**Ethics:**

No ethical concerns

**Limitations:**

Yes the authors have address the limitations in the paper.

**Opportunities For Improvement:**

The authors can add more error analysis to provide insights how and why current MLLMs fail.

**Relation To Prior Work:**

Yes

**Summary And Contributions:**

This paper presents MM-NIAH (Needle In A Multimodal Haystack), a novel benchmark for evaluating multimodal large language models' (MLLMs) ability to comprehend long multimodal documents. The work addresses an important gap in existing MLLM evaluation methods and provides valuable insights into the current capabilities and limitations of state-of-the-art models in this domain.

---

> ### Author Rebuttal · Authors · 2024-08-16
>
> Thank you for your time and expertise in the review process.
>
> **Q1**: The authors can add more error analysis to provide insights how and why current MLLMs fail.
>
> **A1**:
> Thank you for your insightful suggestion.
> We have provided some error analyses in supplementary material (i.e., Appendix A.1 and A.2).
> Please refer to the Common Questions Q2 for details about these analyses.
>
> In summary, based on the current experimental results and analyses, we believe the primary reasons for the model's poor performance include:
> (1) MLLMs are unable to accurately understand images within multimodal documents, and they cannot even distinguish the number of images in a document;
> (2) MLLMs struggle to follow instructions when the context is too long, leading them to produce incoherent or nonsensical outputs;
> (3) The max context length of training samples for existing MLLMs is limited to a relatively short number (e.g., 4096), which is insufficient for models to effectively learn to understand long documents.
>
> We shall include additional error analysis and qualitative examples in the revision.

---

### Official Review · Reviewer_qb5f · 2024-07-25
**New Official Review**

**Rating:** 6
**Confidence:** 4
**Clarity:** Yes.

**Review:**

See my feedback on each section below.

**Strengths:**

1. A timely benchmark for evaluating long-context multimodal understanding ability. One of the first few works that propose the NIAH evaluation for multimodal models.
2. The evaluation is done on both open-source and proprietary models.
3. The paper summarizes several observations from the experiments in Section 4.2, which shares insights into how to improve existing models.

**Additional Feedback:**

No

**Correctness:**

The evaluation and experiment overall make sense to me. One thing that seems a bit questionable is that some models are evaluated on samples exceeding their max context length (Line 204), which makes the accuracy unsurprisingly low.

**Documentation:**

Most details are documented and code is provided. I'd like to see more details about how the question-answer pairs are created.

**Ethics:**

No ethical concerns found.

**Limitations:**

The authors discussed the limitation in the last section, and I agree with them. I think concatenating random samples from OBELICS and just using them as a background is not the same as real-world long-context documents. In addition, I think MM-NIAH seems a bit too challenging for current models (especially open-sourced ones), which makes it difficult to distinguish between the models's long-context understanding ability.

**Opportunities For Improvement:**

1. Some NIAH tasks seem too hard to differentiate between the models, which makes the evaluation less informative. For example, in Figure 3, most models are similarly bad at counting image needles.
2. There are limited details about how the question-answer pairs are generated. Each subset of MM-NIAH has about 2800 questions, how are these questions created and how do they differ from each other?

**Relation To Prior Work:**

Yes.

**Summary And Contributions:**

This paper studies the multimodal (image-text) long-context understanding ability by creating a multimodal needle-in-a-haystack (MM-NIAH) benchmark. MM-NIAH contains two type of “needle” (image and text) and each need type consists of three types of NIAH questions. The benchmark is relatively large-scale, with 12k samples in total. The evaluation on MM-NIAH is conducted on both open-source and proprietary models with a RAG baseline.

---

> ### Author Rebuttal · Authors · 2024-08-16
>
> Thank you for your valuable feedback and constructive comments.
>
> **Q1**: Some NIAH tasks seem too hard to differentiate between the models.
>
> **A1**:
> MM-NIAH includes various tasks and different types of needles, each corresponding to different levels of difficulty. Specifically, Retrieval < Reasoning < Counting, and Text Needle < Image Needle.
> Tasks with text needles provide a good level of differentiation between models (See the first three columns in Figure 3), with closed-source models significantly outperforming InternVL-1.5, which in turn outperforms other open-source models.
> Although all models perform poorly on tasks with image needles, we believe that this highlights a common weakness of current MLLMs, offering direction for future research and provides a platform for further exploration of long multimodal document comprehension.
>
> **Q2**: More details about how the question-answer pairs are generated and how they differ from each other.
>
> **A2**:
> As introduced in Section 3.2, all text needles are generated manually. Image needles for image retrieval and image counting tasks are generated by DALLE-3, while those for image reasoning tasks are sampled from the Jigsaw and Multi-view reasoning split of BLINK benchmark.
> After obtaining the needles, we scatter a randomly sampled needle into a multimodal document to create a evaluation sample in MM-NIAH.
> **Note that the same needle can be scattered into different documents and therefore create different evaluation samples.**
>
> Specifically, we generate 40 different needles for text retrieval, 12 for text counting, 57 for text reasoning, 14 for image retrieval and image counting, and 288 for image reasoning. **We place these needles into different positions within various documents to create different evaluation samples.**
> When generating evaluation samples, we carefully control the distribution of context length and needle depth to ensure they are as uniform as possible.
>
> We shall update these details in the revised version.
>
> **Q3**: Concatenating random samples from OBELICS to serve as a background is not the same as real-world long context documents.
>
> **A3**:
> As of the submission of this paper, there were no open-source multimodal datasets specifically designed for native long-document contexts. At that time, in the available open-source multimodal interleaved datasets, the average document length in MMC4 was 417 tokens, and in OBELICS, it was 816 tokens. So we decided to construct the background document based on OBELICS. We argue that, under the NIAH setting, even by constructing the background through concatenation, many valuable conclusions can be drawn as analyzed in Section 4.2, allowing us to identify some common weaknesses of current MLLMs.
>
> **Q4**: Some models are evaluated on samples exceeding their max context length.
>
> **A4**:
> Thank you for your careful review.
> During evaluation, we directly input the entire document context to MLLMs without any truncation, following the common practice of the long context evaluation in the field of NLP.
> Please refer to the Common Questions Q1 for a detailed response to this question.

---

### Official Review · Reviewer_voQ1 · 2024-07-26
**Presents a benchmark to evaluate three flavors of needle in the haystack style evaluation for both images and text in constructed interleaved documents.**

**Rating:** 5
**Confidence:** 4

**Review:**

Please see below for specifics on quality, clarity, strengths, and weaknesses.

**Strengths:**

* Three notions of needle in the haystack, with progressing difficulty: retrieval, counting, reasoning.
* Many SOTA baselines tested.
* human expert evaluation strengthens the paper

**Additional Feedback:**

N/A

**Clarity:**

The paper flow is reasonable and clear (see Opportunities For Improvement for exceptions).

**Correctness:**

The claims are mostly substantiated; however, key details of the experimental setup remain unclear to me making it hard to fully judge correctness (see Opportunities For Improvement) for specifics.

**Documentation:**

* Documentation exists and there is a github code release.

**Ethics:**

Not at this time.

**Limitations:**

The authors address limitations and mention limited societal impact.

**Opportunities For Improvement:**

* It is not clear why the authors used synthetic images, which may look particularly different from natural document images. Why did the authors not choose image needles for retrieval from OBELICS data directly? Addressing this concern could strengthen the paper.
* Improve document statistics. There can be biases in needle placement. What are the statistics over needle placement over the context length? What about statistics on document lengths in general?
* Point to specific figures in L211
* What is the x-axis in figure 3? I assume this is context length from left to right, but make sure this is clear.
* The paper claims on L211: Performance degrades while context length increases. But do you mean performance degrades as the position of the needle increases in the context length? Is context length fixed at 72k?
* What happens when the document context length is longer than a model's context length?
* Include a plot or figure with text only needle evaluation for models before and after they are multimodal-fine-tuned, e.g., InternLM2 and InternVL-1.5 or similar with the LLaVA family of models.
* It is not clear how RAG is implemented? What model is used as the embedding model for RAG? These are key design and experimental details.
* Consider adding more analysis, figures, and plots. Are there any interesting common failure modes that correlate between models etc. Currently there is only one experimental figure/table.

**Relation To Prior Work:**

* The authors may consider also citing popular image classification and retrieval benchmarks, which were particularly influential for CLIP-style training.

**Summary And Contributions:**

* An evaluation benchmark for multimodal language models on long context documents
* A RAG baseline, which improves long context behavior for text but not images
* Experiments suggesting that existing models (open and closed source weights) struggle with the presented task

---

> ### Author Rebuttal · Authors · 2024-08-16
>
> Thank you for your insightful suggestions and efforts dedicated to the review.
>
> **Q1**: Why use synthetic images?
>
> **A1**:
> We use unrealistic synthetic images to ensure that the inserted image needles are exactly different from the objects existing in the original images within documents.
> In our early experiments, we tried to use image needles from OBELICS data directly but find that such strategy introduces ambiguity.
> For example, when we select a photo of apple from OBELICS data as the image needle, this inserted apple might be confused with the apples already existing in the images, and even human annotators struggle to distinguish them.
>
> **Q2**: Statistics on document lengths and needle placement.
>
> **A2**:
> When generating evaluation samples, we carefully control the distribution of context length and needle depth to ensure they are as uniform as possible.
> Given a fixed context length, we control the distribution of the placement where the inserted needle to ensure it follows a uniform distribution as closely as possible.
>
> Note that the needle depth is defined as $depth = \frac{Needle Position}{Context Length}$. The smaller the needle depth, the closer the inserted needle is to the beginning of the document.
>
> Detailed statistics are shown in Table 1 of the provided pdf, where the horizontal axis represents context length and the vertical axis represents needle depth.
> We shall update these statistics in the revised version.
>
> **Q3**: Point to specific figures in L211
>
> **A3**:
> Thank you for your careful review. Actually, the entire Section 4.2 focuses on Figure 3. The conclusion in L211 is also based on Figure 3. We will clarify this in the revision.
>
> **Q4**: What is the x-axis in Figure 3?
>
> **A4**:
> The x-axis in Figure 3 represents context length, which is consistent with the common practice in NIAH benchmarks[1,2,3,4,5]. We divided the context length into different bins, which form the x-axis of the heatmap.
> For example, the slot in the top left corner represents accuracy when the given context length ranges from 1K to 2K and the needle depth ranges from 0 to 0.2.
> We shall update this in the revision. Thank you for your careful reviews.
>
> **Q5**: Clarify the claim in L211.
>
> **A5**:
> We claim in L211 that the longer the given multimodal documents, the worse the model performs.
> Figure 3 presents the evaluation results in heatmap format.
> Considering the results of InternVL-1.5 on Text Needle Retrieval as an example, for each row in this heatmap, the color of the slots transitions from green to red from left to right, indicating that given a fixed needle depth, performance decreases as the context length increases.
>
> **Q6**: What happens when the context length is longer than a model's context length?
>
> **A6**:
> We directly input the entire document context without any truncation.
> Please refer to the Common Questions Q1 for a detailed response to this question.
>
> **Q7**: Text only needle evaluation for models before and after they are multimodal-finetuned.
>
> **A7**:
> We provide the comparison of InternLM2-20B and InternVL-1.5 in the Table 2-4 of the provided pdf.
> The evaluation of InternLM2-20B is conducted based on their official codebase. We omit the images within the context and only evaluate InternLM2-20B on tasks with text needles.  Note that InternLM2-20B is the language model used to initialize InternVL-1.5.
>
> According to the results in the tables, we can observe that InternLM2-20B and InternVL-1.5 achieve comparable performance when the context length is short. However, when the context length is larger than 32K, the performance of InternVL-1.5 is much inferior to InternLM2-20B, demonstrating that using only samples with a maximum context length of less than 4096 for multimodal fine-tuning can impair the model's ability to handle long contexts.
>
> It is worth noting that InternLM2-20B also performs poorly in counting and reasoning tasks, which are more complex than retrieval. This phenomenon is consistent with the conclusions presented in RULER, which argues that despite achieving nearly perfect performance on the vanilla NIAH test, almost all models exhibit large degradation on more complex tasks as sequence length increases.
>
> We shall update these results and observations into the revised version.
>
> **Q8**: The implementation details of RAG.
>
> **A8**:
> As introduced in L178, we use the text and image encoder of InternVL-G as the embedding model.
>
> To be more concretely, we denote a document as $d=\left(s_1,s_2,i_1,s_3,i_2,s_4,s_5\right)$, where $s_i$ represents a text sentence and $i_i$ represents an image.
> Then we encode $s_i$ and $i_i$ using the text and image encoder of InternVL-G respectively.
> The resulting embeddings are denoted as $e_d=\left(e_{s1},e_{s2},e_{i1},e_{s3},e_{i2},e_{s4},e_{s5}\right)$, where $e_{x}$  represents the embedding of $x$.
> The given question is also encoded by the text encoder of InternVL-G and denoted as $e_q$.
>
> After that, we compute the cosine similarity between $e_q$ and each item in ${e_d}.
> Under the condition that the total token number does not exceed 4096, we retain as many items as possible based on cosine similarity in descending order while preserving their relative order in the document.
> These retained tokens will be inputted into the model.
>
> **Q9**: More analysis about common failure modes that correlate between models etc.
>
> **A9**:
> We provided error analysis in supplementary material (i.e., Appendix A.1 and A.2). Please refer to the Common Questions Q2 for a detailed response to this question.
> We shall add more error analysis and qualitative examples in the revision.
>
> **Q10**: Consider citing popular image classification and retrieval benchmarks.
>
> **A10**:
> We shall add citations about image classification (e.g., ImageNet) and retrieval (e.g., COCO and Flickr30K) benchmarks in the revision.

---

> > ### Comment · Reviewer_voQ1 · 2024-08-30
> > **Rebuttal review**
> >
> > Thanks for the clarification. I plan to up my score to a 6 once the edit functionality is enabled.

---

### Author Rebuttal · Authors · 2024-08-16

Dear all reviewers:

We sincerely appreciate the reviewers for their time and effort in the review. We first address some common questions, followed by detailed responses to each reviewer separately. We hope our responses can clarify existing doubts.

### Common Questions

**Q1**: Some models are evaluated on samples exceeding their max context length.

**A1**:
During evaluation, we directly input the entire document context to MLLMs without any truncation.
We adopt this approach because if we truncate the context to the model's maximum context length used during training, needles inserted into the document might be discarded entirely, making it impossible for the model to answer the given question correctly.

Note that a model's context length generally refers to the maximum sample length during training.
It is possible to input a longer context during inference, provided that it does not exceed hardware limitations (e.g. GPU Memory), although the model's performance may decrease in such cases.

We also emphasize that such evaluation strategy (i.e., input without truncation) is a common practice of the long context evaluation in the field of NLP [1,2,3,4,5,6].

**Q2**: Add more error analysis to provide insights how and why current MLLMs fail.

**A2**:
We have provided error analysis in the supplementary material (i.e., Appendix A.1 and A.2).
Specifically, we observe that MLLMs fail to recognize the exact number of images in the document. As shown in Figure 4 in Appendix, even Gemini, one of the most pioneering MLLMs, struggle to accurately output the number of images in the given context, leading to the inferior performance on image needle tasks of MM-NIAH.
Besides, we find that existing MLLMs are unable to follow instructions and begin to  produce nonsensical text instead of answering the questions when the context length is lengthy (See examples in Figure 5).

In summary, we believe that the reasons for the poor performance of existing MLLMs on MM-NIAH include:
(1) MLLMs are unable to accurately understand images within multimodal documents, and they cannot even distinguish the number of images in a document;
(2) MLLMs struggle to follow instructions when the context is too long, leading them to produce incoherent or nonsensical outputs;
(3) The max context length of training samples for existing MLLMs is limited to a relatively short number (e.g., 4096), which is insufficient for models to effectively learn to understand long documents.

We shall add more error analysis and qualitative examples in the revised version.

**References**

[1] Gregory Kamradt. Needle In A Haystack - pressure testing LLMs. Github, 2023. URL https://github.com/gkamradt/LLMTest_NeedleInAHaystack/tree/main.

[2] Ni, Xuanfan, et al. "XL $^ 2$ Bench: A Benchmark for Extremely Long Context Understanding with Long-range Dependencies." arXiv preprint arXiv:2404.05446 (2024).

[3] Kuratov, Yuri, et al. "In search of needles in a 10m haystack: Recurrent memory finds what llms miss." arXiv preprint arXiv:2402.10790 (2024).

[4] Song, Mingyang, Mao Zheng, and Xuan Luo. "Counting-stars: A simple, efficient, and reasonable strategy for evaluating long-context large language models." arXiv preprint arXiv:2403.11802 (2024).

[5] Hsieh, C. P., et al. (2024). RULER: What's the Real Context Size of Your Long-Context Language Models?. COLM 2024.

[6] Cai, Zheng, et al. "Internlm2 technical report." arXiv preprint arXiv:2403.17297 (2024).

---

### Decision · Program_Chairs · 2024-09-26

**Decision:**

Accept (Poster)

**Comment:**

### Summary
This paper presents MM-NIAH, a new benchmark designed to systematically evaluate the understanding capability of multi-modal models for long multi-modal documents. MM-NIAH employs a "needle in a haystack" approach, hiding secret texts or injecting images (i.e., text and image needles) within long image-text interleaved data. Experiments reveal that current state-of-the-art large multi-modal models struggle to find these needles in long documents, with particularly poor performance on image needles.

### Strengths
- The paper proposes the first "needle in a haystack" dataset to evaluate the comprehension of long documents of multi-modal models.
- The experiments effectively demonstrate the limitations of existing large multi-modal models in handling such tasks.
- The study provides valuable insights into future directions for the field of multi-modal document understanding.

### Weaknesses
- The dataset composition lacks sufficient justification. For instance, the authors insert image tokens after every 2,000 text tokens without adequately explaining this choice or exploring the impact of the ratio for image and text tokens.
- The approach to text needles does not seem to have significantly improved upon the previous text-only NIAH setting.
- The use of synthetic image patches as image needles requires more justification. Alternative approaches, such as using natural image needles, or using rendered text patches with querying in natural language, could be considered.

### Meta-Review
The proposed MM-NIAH benchmark offers a valuable tool for evaluating multi-modal models on long multi-modal documents. The experimental results provide important insights into the current limitations of state-of-the-art models in this task. The Area Chair recommends accepting this paper. Despite some areas needing improvement, the paper presents clear strengths and has significant potential to contribute to the field. Additionally, the authors effectively addressed reviewers' concerns during the rebuttal period.

To further enhance the impact of this research, the authors are encouraged to address the noted weaknesses in the final version. Specifically, providing more detailed justification for dataset composition choices, exploring potential improvements in the text needle approach, and considering alternative methods for image needles would strengthen the paper.